# Systematic Comparison of Uremic Toxin Removal Using Different Hemodialysis Modes: A Single-Center Crossover Prospective Observational Study

**DOI:** 10.3390/biomedicines11020373

**Published:** 2023-01-27

**Authors:** Ariane Duval-Sabatier, Stephane Burtey, Marion Pelletier, Manon Laforet, Laetitia Dou, Marion Sallee, Anne-Marie Lorec, Hafssa Knidiri, Floriane Darbon, Yvon Berland, Philippe Brunet

**Affiliations:** 1Centre de Néphrologie et Transplantation Rénale, Assistance-Publique–Hôpitaux de Marseille, Hôpital de la Conception 147 Bd Baille, 13005 Marseille, France; 2INSERM, C2VN, INRA, Aix Marseille University, 13005 Marseille, France; 3Association des Dialysés Provence et Corse ADPC, 13008 Marseille, France

**Keywords:** dialysis, hemodiafiltration, uremic toxin removal, protein bound solutes, dialysis membrane

## Abstract

Many hypotheses could explain the mortality decrease observed using hemodiafiltration, such as reduction of intradialytic hypotension and more efficient toxin removal. We led a systematic analysis of representative uremic toxin removal with hemodialysis (HD), online postdilution hemodiafiltration (postHDF) and online predilution hemodiafiltration (preHDF), in a single-center crossover and prospective observational study. The primary outcome was the reduction ratio of uremic toxins of the three categories defined by the Eutox group. Twenty-six patients were treated by those three techniques of extra renal epuration. Mean Kt/Vurea was not different between the treatment methods. Mean reduction ratio of beta2microglobulin was significantly higher for both HDF treatments than for HD (*p* < 0.001). Myoglobin, kappa, and lambda free light chain reduction ratio was significantly different between the modes: 37.75 ± 11.95%, 45.31 ± 11% and 61.22 ± 10.56%/57.21 ± 12.5%, 63.53 ± 7.93%, and 68.40 ± 11.79%/29.12 ± 8.44%, 34.73 ± 9.01%, and 45.55 ± 12.31% HD, preHDF, and postHDF, respectively (*p* < 0.001). Mean protein-bound solutes reduction ratio was not different between the different treatments except for PCS with a higher reduction ratio during HDF treatments. Mean albumin loss was always less than 2 g. HDF improved removal of middle molecules but had no effect on indoles concentration without any difference between synthetic dialysis membranes.

## 1. Introduction

Hemodialysis (HD) is a lifesaving treatment. It, moreover, allows nephrologists to manage patients with severe chronic kidney disease before kidney transplantation. However, during the past years, conventional HD has reached a limit of efficiency. Several randomized studies, like the HEMO study, failed to show any benefit of increasing KT/V during HD [1].

Some major therapeutic advances have been made in the field of renal replacement therapy in the last years. Hemodiafiltration reduces mortality compared with conventional hemodialysis [2]. Even if its real biological effects are not yet well understood, many hypotheses could explain the mortality decrease observed using hemodiafiltration, such as reduction of intradialytic hypotension and more efficient toxin removal.

Uremic syndrome is characterized by the retention of various uremic toxins that would normally be excreted by the kidneys and interact negatively with biological functions [3]. According to the classification of uremic retention solutes, these uremic toxins are divided into free water-soluble low-molecular-weight solutes, middle molecules, and protein-bound solutes [4]. Moreover, dialysis aims are not well standardized worldwide. We need to improve this situation by developing and implementing quality metrics for dialysis treatment.

In this single-center prospective observational study, we focused on uremic toxins representative of each classes: Urea for the free water-soluble low-molecular-weight solutes; beta-2-microglobulin (beta2m), myoglobin, kappa free light chain, and lambda free light chain for the middle molecules; p-cresylsulfate (PCS), indole-3-acetic acid (IAA), and indoxyl sulfate (IS) for the protein-bound solutes. We led a systematic analysis of these uremic toxin removal with hemodialysis (HD), online postdilution hemodiafiltration (postHDF), online predilution hemodiafiltration (preHDF).

## 2. Materials and Methods

This is a single-center crossover observational study. The primary endpoint was the comparison of some uremic toxins’ reduction ratio during a session of HD, online pre-HDF, online post-HDF. The secondary endpoint was the comparison of the average albumin loss during a session.

### 2.1. Study Population

The inclusion criteria were patients aged >18 years old with end-stage renal disease from any cause receiving thrice-weekly standard hemodialysis or hemodiafiltration for over 3 months. The exclusion criteria were as follows: Life expectancy less than 6 months, liver cirrhosis, malignancy, albuminemia <30 g/L, need of dialysis session longer than 4 h, single-needle dialysis, and temporary non tunneled catheter. All participants gave their oral informed consent to participate. This study was conducted in our dialysis unit in routine clinical conditions. This work has been approved by our institutional ethic committee.

### 2.2. Treatment Procedures

Patients underwent sessions in each of the three following treatment modes: HD, online pre-HDF, and online post-HDF. Each patient was treated for at least one week using each mode. These modes were successively used in random order. Within each treatment mode, patients could have been treated using up to five different membranes. All these methods were used routinely in our dialysis unit from the beginning of the study. We used synthetic dialysis membranes with high permeability with a surface area of 2.1 m^2^: Cordiax1000HDF (Fresenius Medical Care, Bad Homburg, Germany), Xenium + H21 (Baxter, IL, USA), APS-21H (Asahi Kasei Corporation, Tokyo, Japan), TS-2.1SL (Toray Group, Tokyo, Japan), Elisio-21H (Nipro Corporation, Osaka, Japan). All sessions were conducted on the same generator Nikkiso DBB-05 (Nikkiso, Tokyo, Japan).

Sessions lasted 240 min, the blood flow rate was 300 mL/min, and the dialysate flow rate was 500 mL/min. Hemodialysis anticoagulation was performed with continuous unfractionated heparin infusion at a dose of 25 IU/kg/h during the first hour, 12.5 IU/kg during the second and third hours, and only with patients with a fistula or arteriovenous graft as vascular access, stopped during the last hour. HDF treatment was performed in a volume-control mode using substitution fluid flow rate as a percentage of effective blood flow rate: 80% of the blood flow rate for pre-HDF and 25% for post-HDF, as usual on Nikkiso DBB machines. Both HDF and HD treatments were performed with ultrapure dialysate and online-produced substitution fluid. Substitution volumes were reported as the mean volume obtained in the total of the sessions achieved with each mode.

For this study, we analyzed several toxins concentration in pre- and post-dialysis blood samples. Pre-dialysis sample collection was done just before starting the dialysis session. Post-dialysis sample collection was done at the dialysis session end from the arterial needle after decreasing the blood flow rate to 50 mL/min for 15 s to avoid recirculation [5]. Urea was measured by the spectrophotometric method by UREAL Roche (COBAS 6000 (c501) Roche Diagnostics Basel, Switzerland). Analysis of beta2m was performed by immunonephelemetry with N latex-migroglobulin Siemens (BNPS Siemens Erlangen Germany). Myoglobin concentration was quantified using chimiluminescence done by Advia Centaur XP Siemens (Centaur XP Siemens). Determination of protein-bound solutes concentration was done by HPLC-assay [6]. The reduction ratio (RR) of toxins was defined as a function of pre-dialysis (Cpre) and post-dialysis (Cpost) concentration (RR= (1 − (Cpost/Cpre)) × 100). Concentration at the dialysis end (Cpost) was corrected for hemoconcentration as follows: CCpostCorr = Cpost(1 + Δweight/0.2weight end) [7]. After the sampling of blood, the plasma was extracted and conserved at minus 20 °C before the measurements of various uremic toxins were done. The loss of albumin was measured after total session dialysate collection by immunoturbidimetry (ALBT2 Roche Basel, Switzerland). Each of the values treated is the mean of the values measured for each patient.

### 2.3. Statistical Analysis 

Data are expressed as mean ± standard deviation. Kruskall–Wallis test was used to assess differences among treatments. Test results were considered significant for *p* < 0.05. The R software was used for statistical analysis.

## 3. Results

### 3.1. Baseline Characteristics of the Cohort

Twenty-six patients were included in this study (Table 1). During some sessions, an increase in the transmembrane pressure activates a high alarm and led us to reduce the substitution ratio: 7 in preHDF sessions, 18 in postHDF sessions. We excluded these sessions from the analysis. A total of 166 sessions were analyzed. Mean substitution volumes were 54.79 ± 2.65 L in preHDF, 16.98 ± 1.51 L in postHDF.

### 3.2. Primary Outcome

Mean Kt/Vurea was not different between the different treatment methods: 1.25 ± 0.32 for HD, 1.32 ± 0.28 for pre-HDF, and 1.22 ± 0.33 for post-HDF (*p* = 0.25).

Mean beta2m reduction ratio were 67.06 ± 10.62%, 73.76 ± 6.56%, and 76.66 ± 8.48% for HD, pre-HDF, and post-HDF, respectively (Figure 1). The reduction ratio were significantly higher for both HDF treatments than for HD (*p* < 0.001), but there was no difference between pre-HDF and post-HDF (*p* = 0.34).

Mean myoglobin reduction ratios were 37.75 ± 11.95%, 45.31 ± 11% and 61.22 ± 10.56% for HD, pre-HDF, and post-HDF. All differences between treatments were significant (*p* < 0.001).

Mean kappa free light chain reduction ratios were 57.21 ± 12.5%, 63.53 ± 7.93%, and 68.40 ± 11.79% for HD, pre-HDF, and post-HDF. All differences were significant (*p* < 0.001) (Figure 1). Mean lambda free light chain reduction ratio was 29.12 ± 8.44%, 34.73 ± 9.01%, and 45.55 ± 12.31% for HD, pre-HDF, and post-HDF. All differences were significant (*p* < 0.001).

Mean IAA reduction ratios were 60.14 ± 19.8% for HD, 62.32 ± 12.42% for pre-HDF, and 58.39 ± 18.52% for post-HDF, without any differences between treatments (*p* = 0.48) (Figure 2). Mean IS reduction ratios were 57.17 ± 20.78% for HD, 55.21 ± 12.06% for pre-HDF, and 53.54 ± 13.71% for post-HDF, without any differences between treatments (*p* = 0.84). Mean PCS reduction ratios were 40.04 ± 12.23% for HD, 48.86 ± 12.78% for pre-HDF, and 44.56 ± 11.83% for post-HDF, all differences were significant (*p* < 0.05).

### 3.3. Secondary Outcome, Safety

We also measured albumin loss in the total effluent dialysate during 10 HD sessions, 13 preHDF sessions, and 11 postHDF sessions. Total median albumin loss per dialysis session was 298.2 ± 223.5 mg for HD, 601.5 ± 215.1 mg for preHDF, and 1887.3 ± 1059.1 mg for postHDF (*p* < 0.001).

## 4. Discussion

The main result of this study is the observation of a better elimination of middle molecules with HDF than with HD. Post-HDF had better efficiency than pre-HDF for these toxins. Another important result is the better elimination of the protein-bound uremic toxin PCS with HDF compared to HD. For this toxin, pre-HDF had better efficiency than post-HDF.

The originality of these results is that they gather the data obtained from five different synthetic high-flux dialyzers from different manufacturers: Cordiax1000HDF (Fresenius Medical Care, Bad Homburg, Germany), Xenium + H21 (Baxter, IL, USA), APS-21H (Asahi Kasei Corporation, Japan), TS-2.1SL (Toray Group, Tokyo, Japan), Elisio-21H (Nipro Corporation, Osaka, Japan). They all have the same surface area of 2.1 m^2^. Each dialyzer was used in HD, pre-HDF, and post-HDF. We observed no differences between the five dialyzers. Therefore, the effect observed is strongly related to the techniques, not to the dialyzer properties.

We collected the total effluent dialysate to measure total albumin loss per dialysis session. This method is the gold standard. To our knowledge, none method using sample dialysate has been validated compared with the collection of total dialysates to measure total albumin loss per session.

These results have been obtained under common dialysis conditions. We can, therefore, assume that the data obtained in this study reflect what can be expected under current routine and more general conditions than in other studies.

The substitution volumes used in the present study were 54.79 ± 2.65 L in pre-HDF, and 16.98 ± 1.51 L in post-HDF. These volumes reflect the practice of HDF at the end of the 2000s. It should be noted that this study was carried out before the publication of the ESHOL study in 2013, which shows an improvement in survival obtained with high substitution volumes greater than 20 L in post-HDF [8]. Up to then, randomized controlled trials did not detect a beneficial effect of hemodiafiltration on all-cause mortality [9,10]. It is worth noting that the contrast study explored the convection volume (sum of the intradialytic weight loss plus the substitution volume per session), whereas other studies studied substitution volumes [8,9].

In this study, we thoroughly measured during the same dialysis sessions the reduction ratio of several such representatives of the diversity of the uremic toxins between the three most used renal replacement therapies (HD, pre-HDF, and post-HDF). Comparisons to previous studies calculating RR of uremic retention solutes are quite difficult because of the varying, often considerably differing, treatment parameters and should be analyzed with caution. To interpret our results, we selected studies focused on RR, both for HD and HDF, using high cut-off membranes, with a blood flow rate clearly identified, and biological samples at the end taken from the arterial blood line using the slow-flow method as recommended, as well as hemoconcentration correction.

In 2004, Bammens et al. analyzed the FX80 dialyzer (Fresenius Medical Care, Bad Homburg, Germany) with 14 patients [11]. They reported a better elimination of PCS with HDF compared with HD. They also showed a better elimination of PCS with pre-HDF compared to post-HDF. In the current work, we have obtained similar results concerning PCS with a larger sample size. In 2009, Meert et al. analyzed the Polyflux 170 dialyzer (Gambro, Lund, Sweden) [12]. They reported similar elimination of protein-bound toxins by pre-HDF and post-HDF but did not compare HDF and HD. In 2010, Krieter et al. only compared HD with post-HDF with the PUREMA polyethersulfone membrane (Nipro Corporation, Osaka, Japan) with eight patients [13]. They show a higher removal of beta2m and myoglobin in post-HDF compared to HD (pre-HDF was not assessed in this work). Despite a high substitution volume (>21 L per session in post-HDF), they observed no differences in the elimination of a group of protein-related toxins, including PCS.

Analysis of the efficacy of two polyethersulfone membranes PES-170DS (DIAPES^®^ HF800 membrane) and ELISIO-170H+ from the same producer (Nipro Corporation, Osaka, Japan) in 14 patients showed a better elimination of PCS and IS with pre-HDF and post-HDF compared with HD only with no differences between membranes [14].

A comparison of FLC removal by HD and post-HDF in 31 patients, using FX80 and FX100 dialyzers, was described by Lamy et al. [15]. The substitution volume achieved in post-HDF was around 20 L. They found a difference between post-HDF and HD in kappa FLC removal but no difference in lambda FLC removal. The fact that the elimination of free light chains into dialysate could be predicted by the level of pretreatment plasma concentration could explain the benefit we observed for post-HDF. 

Krieter and colleagues compared IS and PCS removal by low fluxHD, high fluxHD, and post-HDF in a 15-patient series [16]. The sessions were longer than ours (268 ± 17 min), which can explain why they achieved a high substitution volume of 24.3 L in post-HDF. Pre-HDF was not studied in this work. They showed a better total IS and PCS removal in post-HDF compared to HD but failed to show an impact on IS and PCS plasma predialysis levels after six weeks.

In order to increase the free fraction of the protein-bound toxins, some other approaches have been tried [17], like using binding competitor during dialysis [18], oral or plasma adsorbent. Further research is needed in this field.

We observed in this work that total albumin loss is significantly higher in both pre-HDF and post-HDF compared to HD and achieves nearly 2 g per session in post-HDF. We observed lower albumin loss compared with older studies. In 2004, Ahrenholz et al. observed total albumin loss in total effluent dialysate up to 7 gr per session [19]. This could be due to the improvement of selectivity in the new generation of membranes used in our study.

Our study has some limitations. Because of the middle size of this cohort, we could not study the influence of substitution volumes on the toxin’s RRs. The study duration was also too short to allow a comprehensive analysis of pre-dialysis toxins plasma levels. In 2010, Meert et al. showed a significant decrease in pre-dialysis concentration of total p-cresyl sulfate when measured nine weeks after switching from HD to post-HDF [20].

Regardless of uremic toxin removal, chronic kidney disease causes dysregulation of the production of these toxins, of which the gut microbiome is a central actor. Intestinal dysbiosis due to uremia and the decreased fiber intake due to a very restrictive diet lead to the proliferation of proteolytic bacteria. These bacteria generate excess amounts of potentially toxic compounds. Impaired intestinal barrier function permits translocation of gut-derived uremic toxins into the systemic circulation. Work on this gut–kidney axis opens up a new field of complementary therapies to dialysis [21].

## 5. Conclusions

We compared, in a systematic manner, the removal of most uremic retention solutes with different treatment modes. We confirmed that HDF improved the removal of middle molecules and PCS had no effect on urea and indoles epuration. Our study confirmed the benefits of HDF on PCS elimination. Our study provides a strong rationale to suggest PCS removal as an interesting biomarker to estimate the quality of HDF besides middle molecules. We need to confirm these results in larger cohorts and longer follow-ups to improve our knowledge about the gain of survival observed with convective therapies: Due to better removal of uremic solutes, or due to other factors, like hemodynamic stability improvement?

## Figures and Tables

**Figure 1 biomedicines-11-00373-f001:**
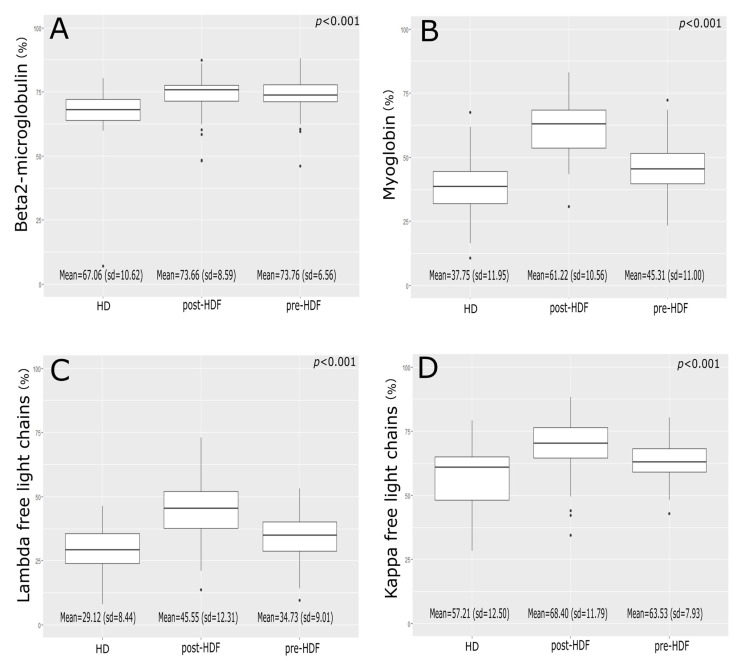
Middle molecules reduction ratio. (**A**) β2µglobulin (beta2m), (**B**) Myoglobin, (**C**) Lambda free light chains, (**D**) Lambda free light chains. HD: Hemodialysis; HDF: Hemodiafiltration; sd: Standard deviation.

**Figure 2 biomedicines-11-00373-f002:**
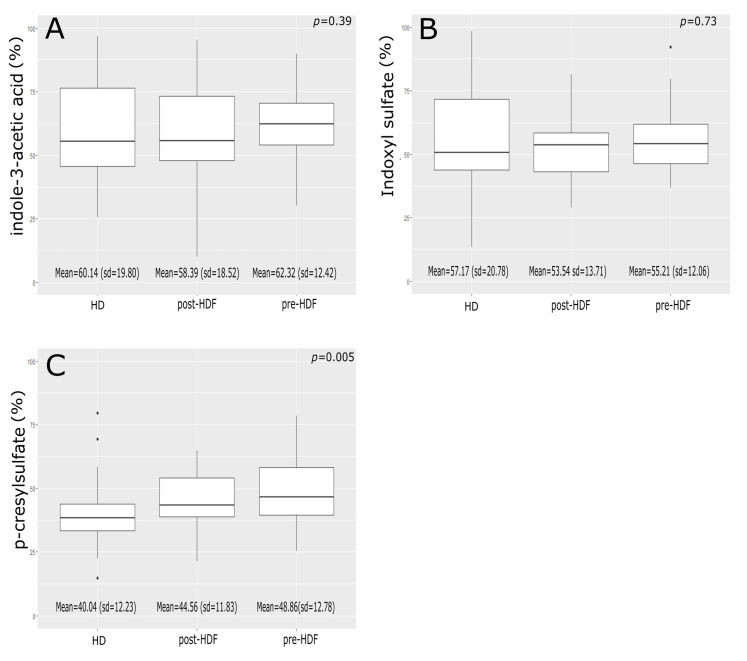
Protein-bound solutes reduction ratio. (**A**) Indole-3-Acetic acid, (**B**) Indoxyl Sulfate, (**C**) *p*-Cresyl Sulfate. HD: Hemodialysis; HDF: Hemodiafiltration; sd: Standard deviation.

**Table 1 biomedicines-11-00373-t001:** Patients’ characteristics.

		N = 26 Patients	
**Age (year)**		72.50 (26–87)	
**Male sex**		17 (65.4%)	
**Diabetes**		6 (23%)	
**Cardiovascular disease**		16 (61.5%)	
**Primary renal disease**		Vascular	9
		Diabetes	6
		PKAD	3
		Glomerulonephritis	4
		Other/Unknown	4
**Hemoglobin level**		110.4 g/L (±16.9)	
**Albumin level**		36.6 g/L (±3.4)	
**Time on dialysis (months)**		19 (4–117)	
**Modified Charlson index**		9 (3–13)	
**Vascular access**	**Arterio-venous Fistula**	22 (84.6%)	
**Catheter**		4 (15.4%)	
**Anuric patients**		14 (53.8%)	

Data are presented as **N** (%) or median (min-max).

## Data Availability

Data are available directly from the first author.

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
