# Peer review of "Systematic Comparison of Uremic Toxin Removal Using Different Hemodialysis Modes: A Single-Center Crossover Prospective Observational Study"

_biomedicines, 2023, doi:10.3390/biomedicines11020373_

Round 1

Reviewer 1 Report

Comments and Suggestions for Authors:

The authors submitted a research article in which they evaluated a significance of removing uremic toxins with hemodialysis (HD), on-line postdilution hemodiafiltration (postHDF), on-line predilution hemodiafiltration in single-center cross-over observational study. 26 patients underwent sessions in each of the three following treatment modes: HD, on-line preHDF, and on-line postHDF (7 patients were included in preHDF sessions; 18 individuals were allocated in postHDF sessions). The authors evaluated several toxins concentration in pre- and post-dialysis blood samples and found that . The strength of the article is a hot topic of the stuty. The weaknesses are a lack of control group and small sample size. However, the finding of the study seem to be impessive, so I would like to put forward several comments to discuss

1.      The authors should more thoroughly describe how they estimated ample size.

2.      Flow chart with clear inclusion / exclusion criteria are required to be reporte.

3.      Basic characteristics of the patients should include clinical signes / symptoms, comorbidities, concomitant diseases, risk factors, biochemistry, but not uremic toxibs levels and methods of their removing.

4.      Description of blood sample evaluation is missed. Please, add this information.

5.      Body mass index dry mass should be evaluated along with given in the aricle Kt/v

6.      Descritive statistics is not completely corresponded to the aim of the study. Please, check it and add log regression

Author Response

We thank the reviewers for these interesting comments. All patients the 26 patients corresponding to inclusion criteria were included and performed each technic of dialysis. We did not allocated subsample of patients to different technics. We only excluded 7 sessions in preHDF and 18 sessions in post HDF from the analysis regarding alarming. This was already explain in the first paragraph of the results.

  1. The authors should more thoroughly describe how they estimated ample size.

We did not perform sample size calculation because this study was conducted in our dialysis unit in routine clinical conditions.

  1. Flow chart with clear inclusion / exclusion criteria are required to be reporte.

The inclusion criteria are clearly described in the methods. All the 26 patients performed the various technic without allocations for various technic so the flow chart did not provide additional informations.

  1. Basic characteristics of the patients should include clinical signes / symptoms, comorbidities, concomitant diseases, risk factors, biochemistry, but not uremic toxibs levels and methods of their removing.

We added information in table 1 regarding the comorbidities (cardiovascular disease, kidney disease) and basic biochemistry (albumin and Hemoglobin concentration) relevant to study

  1. Description of blood sample evaluation is missed. Please, add this information.

After the sampling of blood, the plasma was extracted and conserved at minus 20°C before the measurements of various uremic toxins were done.

  1. Body mass index dry mass should be evaluated along with given in the aricle Kt/v

We did not have the body mass index dry mass.

  1. Descritive statistics is not completely corresponded to the aim of the study. Please, check it and add log regression

We performed descriptive analysis and a statistical test adapted to the data Kruskall-Wallis test, our statistician told us that logistic regression is not adapted to this study.

Reviewer 2 Report

In this interesting , well designed study in current practice, authors compare extraction of urea,  middle MW uremic toxins and protein-bound uremic toxins in dialysis sessions performed with traditional hemodialysis technic , predilution hemodialfitration (PreHDF) and post-dilution  hemodialfitration (Post HDF)and elegantly show the superiority of PreHDF and Post HDF for removal of middle MW uremic toxins and P-cresylsulfate a protein bound toxin. These informations are new since most studies either focused on removal of middle MW uremic toxins upon different dialyzers generally during a dialysis single session(studies by Pottier et al and by Manuel et al) or did not compare such a panel of uremic toxins with these 3 hemodialysis technics.

Minor comments :

Table 1 : Charlson : I guess that it is the "modified Charlson index" (including age) that the authors used

Summary

Authors stated That " "Mean protein-bound solutes reduction ratio was not different between the different treatment."whereas they demonstrated that P-cresyl sulfate level was better reduced by Predilution HDF and Postdilution HDF; therefore the sentence in the summary should be modified accordingly.

Author Response

We thank the reviewers for its nice comment.

Table 1 : Charlson : I guess that it is the "modified Charlson index" (including age) that the authors used

We corrected it in the table 1

Summary

Authors stated That " "Mean protein-bound solutes reduction ratio was not different between the different treatment."whereas they demonstrated that P-cresyl sulfate level was better reduced by Predilution HDF and Postdilution HDF; therefore the sentence in the summary should be modified accordingly.

We corrected the sentence in the abstract

Reviewer 3 Report

The conclusions of your study highlight facts already established by many studies performed over the years. Therefore, it is hard to determine the novelty of your study regarding the differences between these types of renal replacement therapy. In addition, there are recent studies that already determined the influence of different synthetic dialyzers (high and low-flux) on efficiently removing different types of molecules. Furthermore, your study does not present the medical history of your patients (primary renal disease, treatment, concomitent associated pathologies and complications associated to renal impairment, such as the degree of anemia, CKD-MBD etc.) and parameters related to the RRT session (i.e., anticoagulation) that might influence the final results.

Author Response

We thank the reviewer for its interesting comments. In method we added a sentence regarding the anticoagulation

"Hemodialysis anticoagulation was performed with continuous unfractionated heparin infusion at a dose of 25 IU/kg/h during the first hour, 12.5 IU/kg during the second and third hours, and, only with patient with a fistula or arterioveinous graft as vascular access, stop during the last hour)."

We added supplementary information in table 1 regarding the comorbidities (cardiovascular disease, kidney disease) and basic biochemistry (albumin and Hemoglobin concentration) relevant to study

Round 2

Reviewer 3 Report

Even if you added some new data, the study does not provide any new information regarding these different forms of renal replacement therapy. The results and conclusions have been already well documented in the international literature.